# Does Health Insurance Modify the Association Between Race and Cancer-Specific Survival in Patients with Urinary Bladder Malignancy in the U.S.?

**DOI:** 10.3390/ijerph16183393

**Published:** 2019-09-13

**Authors:** Juliana Morales, Aaron Malles, Marrell Kimble, Pura Rodriguez de la Vega, Grettel Castro, Alan M. Nieder, Noël C. Barengo

**Affiliations:** 1Herbert Wertheim College of Medicine, Florida International University, 11200 SW 8th Street, Miami, FL 33199, USA; jmora075@med.fiu.edu (J.M.); amall018@med.fiu.edu (A.M.); mkimb011@med.fiu.edu (M.K.); 2Department of Medical and Population Health Sciences Research, Herbert Wertheim College of Medicine, Florida International University, 11200 SW 8th Street, Miami, FL 33199, USA; rodrigup@fiu.edu (P.R.d.l.V.); gcastro@fiu.edu (G.C.); 3Department of Urology, Herbert Wertheim College of Medicine, Florida International University, 11200 SW 8th Street, Miami, FL 33199, USA; Alan.Nieder@msmc.com; 4Faculty of Medicine, Rīga Stradiņš University, LV-1007 Riga, Latvia; 5Department of Public Health, Faculty of Medicine, University of Helsinki, 00014 Helsinki, Finland

**Keywords:** carcinoma, transitional cell, mortality, survival rate, urinary bladder neoplasms

## Abstract

*Background*: Scientific evidence on the effect of health insurance on racial disparities in urinary bladder cancer patients’ survival is scant. The objective of our study was to determine whether insurance status modifies the association between race and bladder cancer specific survival during 2007–2015. *Methods*: The 2015 database of the cancer surveillance program of the National Cancer Institute (*n* = 39,587) was used. The independent variable was race (White, Black and Asian Pacific Islanders (API)), the main outcome was cancer specific survival. Health insurance was divided into uninsured, any Medicaid and insured. An adjusted model with an interaction term for race and insurance status was computed. Unadjusted and adjusted Cox regression analysis were applied. *Results*: Health insurance was a statistically significant effect modifier of the association between race and survival. Whereas, API had a lower hazard of death among the patients with Medicaid insurance (HR 0.67; 95% CI 0.48–0.94 compared with White patients, no differences in survival was found between Black and White urinary bladder carcinoma patients (HR 1.24; 95% CI 0.95–1.61). This may be due a lack of power. Among the insured study participants, Blacks were 1.46 times more likely than Whites to die of bladder cancer during the 5-year follow-up (95% CI 1.30–1.64). *Conclusions*: While race is accepted as a poor prognostic factor in the mortality from bladder cancer, insurance status can help to explain some of the survival differences across races.

## 1. Introduction

The American Cancer Society estimated that in 2018 there will be 81,190 new cases of bladder cancer with an estimated number of new deaths projected to be 17,240, making it the fifth most common cause of cancer in the U.S. [1]. From the years 1989 to 2013, the five year survival have remained relatively stable [1]. 

The current scientific evidence on the association between race and overall survival of patients with bladder cancer consistently reports a disparity between blacks and Whites [1,2,3,4,5,6,7,8,9,10,11,12,13,14]. Retrospective cohort studies using data from the Surveillance, Epidemiology, and End Results Program (SEER) database and National Cancer Databases (NCDB) have found that the five year overall survival from bladder cancer in Blacks was significantly lower than the overall survival of Whites during the last three decades [2,3,4,8,15]. 

Whereas most of the previous studies controlled their statistical models for marital status, gender, tumor-related characteristics types of surgeries, other clinical factors, only very few of them tried to assess the associations between health insurance or whether the association between race and survival depends on the health insurance of the patients [6,12]. Given that in 2017, the census bureau found that of a total of 323,156 participants 9% of the population was uninsured (*n* = 28,543), information on survival according to health insurance in urinary bladder cancer patients may help to address the needs of the most vulnerable population groups [16]. Finally, information in regard association between race, insurance and survival in Asian/Pacific Islanders is scant.

Therefore, the aim of this study was to determine whether insurance status modifies the association between race and bladder cancer-specific survival in patients with urinary bladder malignancy in the U.S. We hypothesize that race as a prognostic factor for 5-year bladder cancer specific survival behaves differently depending on the insurance status of the patient.

## 2. Materials and Methods 

### 2.1. Study Design and Setting

A secondary data analysis of the Surveillance Epidemiology and End Results (SEER 9) database was conducted. SEER is a cancer surveillance program of the National Cancer Institute’s (NCI) that collects cancer incidence and survival data since 1973, including the Atlanta, Detroit, San Francisco Oakland, Seattle-Puget Sound metropolitan areas and the states of Connecticut, Hawaii, Iowa, New Mexico, and Utah [17]. 

### 2.2. Study Participants

The population of interest included adults 18 years-of-age or older, identified as White, Black, and Asian Pacific Islander (API) with a first-time diagnosis of urinary bladder malignancy (International Classification of Diseases for Oncology, 3rd Edition [ICD-O-3], codes C67.0-C67.9, and histology code 8120-8130 from 2007 to 2015 (*n* = 39,587). Thus, the study population was restricted to first primary and histologically confirmed cases. The period of this study was chosen due to the SEER 9 database beginning to record insurance status in 2007. Cases with missing data on race (*n* = 504), unknown cause of death or missing information on survival (*n* = 1071), confirmed or possible diagnosis at autopsy (*n* = 171) or duplicate cases (*n* = 176) were excluded. For the rest of the variables in the study, the percentage of missing was less than 8% (status: 7.97%, insurance: 5.30%, stage: 1.23%, surgery: 0.18%). The final population consisted of 37,871 patients.

### 2.3. Main Variables

The primary outcome was five-year cause-specific mortality which was defined as the time in months from diagnosis to death due to cancer specific causes. Patients who were alive at the date of the last contact (60 months after diagnosis) were censored. Race was defined as White, Black, and API. Healthcare coverage in the U.S. is provided through a combination of private health insurance and public health coverage such as Medicaid. It has been estimated that close to 91% percent of the U.S. population had some type of health insurance. Most Americans with private health insurance receive it through an employer-sponsored program. Medicaid in the United States is a federal and state program that helps with medical costs for some people with limited income and resources. People who do not have a health-insurance (the uninsured) have to pay for their health services. In this study, health insurance status was grouped into uninsured, any Medicaid and Insured (combining SEER Insured and Insured/No specifics categories). SEER report insurance status of the patient based on the primary payer at diagnosis according to NAACCR Field Primary Payer at diagnosis. SEER does not provide much detail about the services covered by Medicaid or other insurances. 

Other covariates included age at diagnosis, gender, ethnicity, marital status, grade at diagnosis, stage at diagnosis, and surgical treatment. Age in years was divided into the following categories <50, 50–59, 60–69, 70–79, 80–89, 90+ years-of-age. Ethnicity was dichotomized into Hispanic or non-Hispanic. Marital status was defined as partnered (married, and unmarried/domestic partner) and un-partnered (single, separated, divorced, and widowed). Grade at diagnosis was categorized in four groups: (1) well or moderately differentiated, (2) poorly differentiated, (3) undifferentiated, and (4) ungraded. Stage at diagnosis categories was defined as localized, regional, and distant; patients that were un-staged at diagnosis were excluded. Surgical treatment was dichotomized into surgical treatment not performed or performed (this category included codes for local tumor destruction not otherwise specified (NOS), local tumor excision NOS, electrocautery, excisional biopsy (TURB), complete cystectomy with reconstruction, radical cystectomy plus ileac conduit, radical cystectomy including anterior exenteration.

### 2.4. Statistical Methods

Statistical analysis was performed using STATA (StataCorp LP., College Station, TX, USA) [18]. Categorical variables are presented as proportions. Chi-square tests were used to compare categorical covariates according to the exposure and outcome variables. Kaplan-Meier survival analysis was used to compare overall survival curves. The log-rank test was used to assess differences between survival curves across race/ethnicity. Collinearity was assessed via pairwise correlations. A Pearson’s correlation coefficient absolute value of 0.67 or higher was used for identifying collinearity among a pair of variables. In the study, no collinearity was found among any of the variables included. Patients were follow-up for a maximum length of five years. Unadjusted and adjusted Cox Proportional hazard regression models were used to calculate hazard ratios (HRs) and their corresponding 95% confidence intervals (CIs). Age at diagnosis, gender, marital status at diagnosis, grade at diagnosis, stage at diagnosis, and surgical treatment were included in the adjusted model. Wald test was performed for each variable in order to test the statistical significance of each variable in the model. The adjusted model with an interaction term for race and insurance status was fitted. As at least one of the interaction terms of race*insurance was statistically significant, showing possible effect modification by health insurance, the adjusted model was stratified for each stratum of health insurance. The proportional hazard assumptions were tested graphically. All *p* values reported are two-tailed, and a *p*-value of <0.05 was considered as statistically significant. 

### 2.5. Ethical Considerations

Permission to use and access to the SEER database was obtained through the SEER website. All data accessed from SEER was de-identified (fully anonymized) and without any of the 18 direct identifiers according to the Health Insurance Portability and Accountability Act. Ethical approval was waived by the Florida International University Health Sciences IRB since the analysis was considered nonhuman subjects research using de-identified data.

## 3. Results

Table 1 describes the demographic and baseline clinical characteristics of the study participants diagnosed with bladder cancer in the SEER database between 2007 and 2015. Across all races, a higher proportion of patients were diagnosed with bladder cancer between the ages of 60–69 years (White 28%, Black 31.2%, API 26.2%) and 70–79 years (White 28.8%, Black 22.5%, API 28.2%). Furthermore, a higher proportion of Blacks were diagnosed with bladder cancer at a younger age (<50, 50–59, 60–69) compared with Whites and API. Blacks also had a higher frequency of being females (32.7%), unpartnered (58.7%), and being either uninsured (3.7%) or having Medicaid (13.0%) compared with Whites and API. Regional (22.2%) or distant (7.03%) cancer bladder cancer stage as well as higher grading of the disease were more prevalent in Blacks than in the Whites and API study population. Blacks also had a lower prevalence of patients who underwent surgery (91.1%) than Whites and API. All of these differences in demographic and clinical characteristics between the race groups were found to be statistically significant (*p*-values < 0.001).

Figure 1, Figure 2 and Figure 3 show the Kaplan Meier curves for cancer specific survival in months according to health insurance. The log-rank test revealed a statistically significant difference in survival among the different races in the insured patients and those with Medicaid (*p*-values < 0.001). Black had lower survival compared with White or API in both health insurance groups. However, no differences in survival was found between White, Black or API uninsured patients (Figure 3; *p*-value = 0.429).

The range of follow up was between 0 and 60 months. Median survival time could not be computed since more than 50% of individuals were alive at the end of follow up. The mean survival time for the sample was 52.6 months. Table 2 presents the adjusted hazard ratios for cause-specific 5-year survival with the interaction term Race*Insurance included. The interaction term API & Any Medicaid was statistically significant suggesting possible interaction between health-insurance and survival.

The interaction of race*insurance was statistically significant in the adjusted logistic regression models revealing possible effect modification by health insurance (data not shown). Therefore, the adjusted model was stratified according to health insurance. The adjusted hazard ratios for cancer-specific survival stratified by insurance are shown in Table 2. There was no statistically significant association between race and survival between White, Blacks, API urinary bladder cancer patients that were uninsured (HR for Black 1.19 [95% CI 0.58–2.42); HR for API 0.44 [95% CI 0.10–1.89]). Whereas, API had a lower hazard of death among the patients with Medicaid insurance (HR 0.67; 95% CI 0.48–0.94 compared with White patients, no differences in survival was found between Black and White urinary bladder carcinoma patients (HR 1.24; 95% CI 0.95–1.61). Among the insured study participants, Blacks were 1.46 times more likely than Whites to die of bladder cancer during the 5-year follow-up (95% CI 1.30–1.64). However, insured API had the same survival than their White counterparts (HR 0.99; 95% CI 0.86–1.14). As the most important secondary finding, it may be noteworthy to mention that among the insured patients, Hispanics were more likely to die from bladder cancer than non-Hispanics (HR 1.21; 95% CI 1.02–1.43). 

## 4. Discussion

Our study showed that the association between race and 5-year survival in bladder cancer patients depended on insurance status. Black patients who were insured had a higher hazard of death compared with their White counterparts, whereas being API, the survival is better in the Medicaid patient group. Hispanic patients with health insurance had a higher hazard of death compared with non-Hispanics. Finally, the post-hoc power calculations and small sample size of the uninsured in the sample revealed a low power to detect a statistically significant results when comparing Black or API with White individuals among uninsured patients. Thus, one of the reasons why our data did not reveal a statistically significant associations between race and survival among the Black uninsured patients and those with Medicaid may be due to lack of power.

Our findings revealed that the association between race and survival depends on the insurance status of the patient. Our data is in line with the previous scientific evidence [6,12]. Mallin et al. reported that bladder transitional cell carcinoma patients who were either uninsured or had Medicaid—were at 50%, respectively 70% increased risks of death compared with patients who were privately insured [6]. A study examining the association between health insurance status and survival of New Jersey patients 18–64 diagnosed with seven common cancers during 1999–2004 found that uninsured bladder cancer patients had statistically significantly lower survival rates than privately insured patients [12]. Uninsured patients were 76% more likely to die from bladder cancer compared to insured patients, respectively [12]. On contrast to our study, previous ones did not report findings including API. A retrospective analysis of the cancer stage at diagnosis for 12 cancer sites in the USA, diagnosed between 1998 and 2004 using the US National Cancer Database reported that uninsured and Medicaid-insured patients are at least twice as likely to present with regional disease and 60% more likely to have locally-advanced disease at diagnosis compared with privately insured patients [10,13]. A possible non-biological explanation for the survival disparity according to health insurance is the documented inequality Blacks face in access to quality care. Blacks who are insured may be restricted to community-based healthcare systems that may not have the experience or infrastructure to care for complex conditions [10,19,20]. Other factors such as transportation, affordability, and distrust of the healthcare system have historically hindered Blacks from receiving timely care [10,21]. Insurance status is not the only factor that contributes to a patient’s access to care. Transportation, willingness to seek medical care, attitudes toward healthcare providers among other factors may also play a role [10]. 

Our data is in agreement with most of the few previous studies assessing survival in urinary bladder cancer patients according to race [3,5,6,7,22]. A study in U.S. urinary bladder cancer patients using data from SEER during 1973 and 2014 found that Non-Hispanic Whites had the best survival rates during a five-year follow-up [7]. Moreover, they also reported that Blacks had a statistically significant lower survival than other racial groups even when controlling for gender, age, histology, tumor size and surgical treatment [7]. Another study following-up patients diagnosed with transitional cell carcinoma of the bladder between 1975 and 2005, reported that cancer-specific survival was reduced in blacks compared with other ethnic groups, even when stratified by stage and grade [3]. Also, data from the Florida Cancer Data System and the Agency for Health Care Administration data sets (1998–2003) revealed a statistically significantly longer survival in Whites (63.0 months) compared with Blacks (39.6 months) [5]. However, a recent analysis of overall survival and bladder cancer-specific survival in patients with metastatic bladder cancer using the SEER 2010–2013 data did not find any statistically significant differences in survival between Whites and Blacks [22]. Moreover, an analysis of bladder transitional cell carcinoma cases diagnosed in 1993 to 2007 from the National Cancer Data Base found that the 30% higher mortality risk in black males and females was primarily limited to late stage diseases [6].

Several factors may explain why Blacks have a decreased survival compared with White bladder cancer patients. Possible explanations for this disparity are differences in the smoking prevalence, comorbid conditions, type and quality of care, genetics and socioeconomic status between Blacks and Whites [6,9,10,20]. It has also been suggested that the racial difference in cancer-specific survival may be due to the advance stage of disease at time of diagnostics [6,9,10]. Gild et al. found a difference in length of time to neoadjuvant chemotherapy and radical cystectomy, amount of lymph nodes removed, length of stay in days, neoadjuvant chemotherapy utilization, and pelvic lymph node dissection between Blacks and Whites [4]. They concluded that Black patients with muscle invasive bladder cancer undergoing radical cystectomy and possible neoadjuvant chemotherapy may receive a lower quality of care than White patients. Finally, factors related to health-care provider or accessing to medical care or differential treatment of patients by race may play an important role in explaining the existing health disparity between Blacks and Whites. Thus, further studies may explore in more depth socioeconomic barriers such as income, factors related to health education or medical depths between different race to access health-care and proper treatment and surveillance [23].

Naturally, our study had some limitations. The SEER database does not record detailed information about significant lifestyle habits such as smoking status, occupational exposures and educational status. Also, information on genetic factors were not available. Thus, these factors may over or underestimate the strength of the association between race, health insurance and survival. Moreover, SEER assumes all patients age > 65 years have Medicare, but codes them as insured, making it impossible to further stratify insurance status from public or private sources which we feel could have highlighted further disparities. Furthermore, it was not possible to identify the primary site of care of patients, thus access to appropriate care could have affected our findings. Finally, we were also limited in the information available about surgical treatments performed since only the initial surgical procedure was recorded rather than the most definitive treatment, making it impossible to delineate differences in level of care received (e.g., number of lymph nodes sampled, surgical procedures performed/offered or skills of the surgeons). 

## 5. Conclusions

In conclusion, our results indicated that while race is largely accepted as a poor prognostic factor in bladder cancer specific survival, insurance status can help to explain some of the prognostic differences across races, though it does not fully elucidate the cause of the disparity. Differences in mortality from bladder cancer is multifactorial and is associated with quality of care and access to treatment. This may mean that in addition to the biologic differences between races, insurance status plays a pivotal role in the cancer survival. In light of these results, we suggest to include interventions for clinicians to identify high risk patient populations and connecting them with additional services such as educational materials, case management, transportation services, and identification of other social determinants of health that impede access to quality care. Future efforts should be made to further explore the variation in access to care, quality of care, and institutional availability for patients of varying insurance status to improve health outcomes and minimize disparities in cancer survival between races for patients with bladder cancer.

## Figures and Tables

**Figure 1 ijerph-16-03393-f001:**
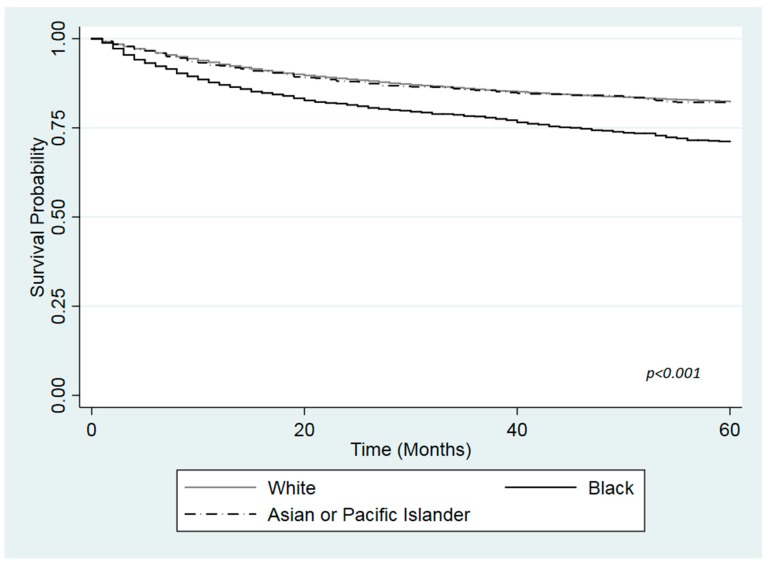
Kaplan-Meier curve for cause specific survival by race among insured patients with urinary bladder cancer, based on the Surveillance, Epidemiology, and End Results Program (SEER) cohort, 2007–2015.

**Figure 2 ijerph-16-03393-f002:**
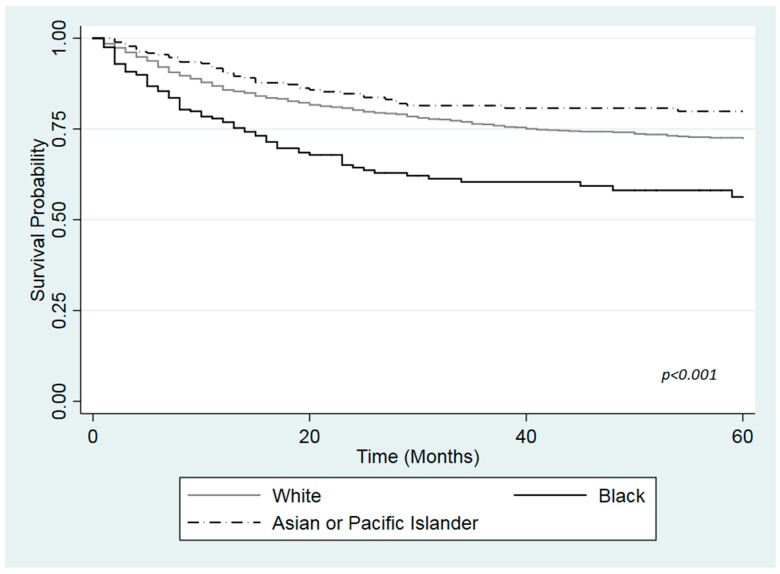
Kaplan-Meier curve for cause specific survival by race among Medicaid patients with urinary bladder cancer, based on the SEER cohort, 2007–2015.

**Figure 3 ijerph-16-03393-f003:**
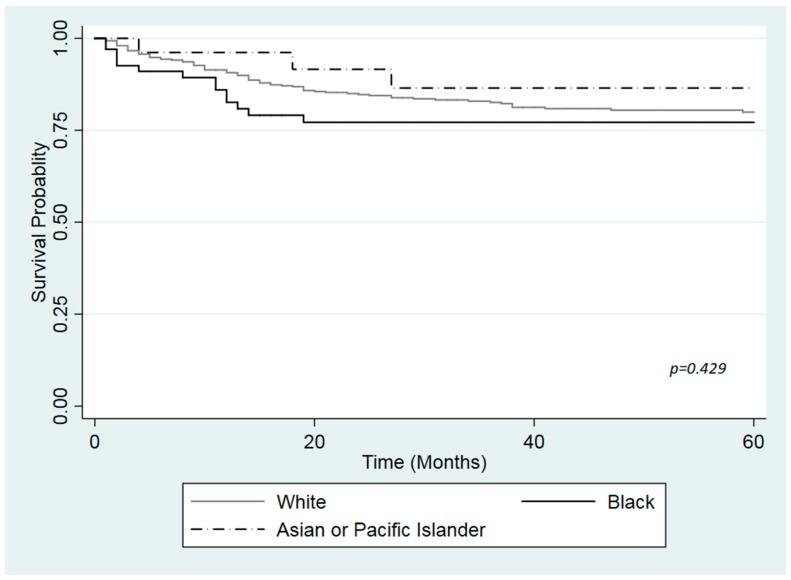
Kaplan-Meier curve for cause specific survival by race among uninsured patients with urinary bladder cancer, based on the SEER cohort, 2007–2015.

**Table 1 ijerph-16-03393-t001:** Baseline characteristics of study participants with primary urinary bladder cancer according to race during 2007–2015.

Characteristics	Race			*p*-Value ^2^
	White (*N* = 33,876)	Black (*N* = 2005)	API ^1^ (*N* = 1990)	
	%	%	%	
Age (years)				<0.001
<50	4.78	7.73	4.37	
50–59	14.3	20.85	13.57	
60–69	27.95	31.22	26.18	
70–79	28.75	22.54	28.19	
80–89	20.51	14.71	22.76	
90+	3.21	2.94	4.92	
Gender				<0.001
Male	77.31	67.28	74.87	
Female	22.69	32.72	25.13	
Ethnicity				<0.001
Non-Hispanic	96.57	99.2	98.49	
Hispanic	4.43	0.8	1.51	
Marital status				<0.001
Partnered	64.93	41.31	68.78	
Unpartnered	35.05	56.69	31.22	
Insurance status				<0.001
Uninsured	1.47	3.68	1.58	
Any Medicaid	5.52	13.02	15.05	
Insurance	93.02	83.3	83.37	
Stage				<0.001
Localized	79.78	70.82	77.82	
Regional	16.92	22.15	18.7	
Distant	3.3	7.03	3.47	
Grade				<0.001
Grade I/II	32.98	24.44	33.07	
Grade III	13.24	15.62	11.61	
Grade IV	34.06	38.65	41.46	
Not Graded	19.72	21.75	13.87	
Surgery				<0.001
Surgery	94.52	91.05	95.09	
No Surgery	5.48	0.95	4.91	

^1^ Asian/Pacific Islanders; ^2^ Chi-square test.

**Table 2 ijerph-16-03393-t002:** Adjusted hazard ratios for cause-specific 5-year survival stratified by insurance status among adult patients diagnosed with bladder cancer (SEER 2007–2015).

Characteristics	Adjusted Estimates					
	Uninsured		Medicaid		Insured	
	(*n* = 571)	Wald Test *p*-Values	(*n* = 2304)	Wald Test *p*-Values	(*n* = 32,988)	Wald Test *p*-Values
	HR ^a^ (95% CI ^b^)		HR (95% CI)		HR (95% CI)	
Race		0.471		0.013		<0.001
White	Ref. ^c^		Ref.		Ref.	
Black	1.19 (0.58–2.42)		1.24 (0.95–1.61)		1.46 (1.30–1.64)	
API ^d^	0.44 (0.10–1.89)		0.67 (0.48–0.94)		0.99 (0.86–1.14)	
Age (years)		0.002		<0.001		<0.001
<50	Ref.		Ref.		Ref.	
50–59	1.23 (0.56–2.70)		1.24 (0.83–1.85)		1.22 (0.97–1.55)	
60–69	1.19 (0.54–2.62)		1.18 (0.80–1.75)		1.50 (1.20–1.87)	
70–79	1.42 (0.45–4.49)		1.48 (0.98–2.22)		2.03 (1.63–2.53)	
80–89	2.06 (0.59–7.22)		2.29 (1.50–3.49)		3.52 (2.83–4.38)	
90+	17.40 (4.28–70.65)		2.61 (1.29–5.29)		6.52 (5.13–8.27)	
Gender		0.888		0.176		0.058
Male	Ref.		Ref.		Ref.	
Female	1.04 (0.59–1.83)		1.16 (0.94–1.44)		1.07 (1.00–1.15)	
Ethnicity		0.936		0.027		0.025
Non-Hispanic	Ref.		Ref.		Ref.	
Hispanic	1.03 (0.42–2.53)		0.68 (0.49–0.96)		1.21 (1.02–1.43)	
Marital Status		0.966		0.121		<0.001
Partnered	Ref.		Ref.		Ref.	
Unpartnered	0.99 (0.62-1.59)		1.19 (0.96–1.48)		1.29 (1.21–1.38)	
Stage		<0.001		<0.001		<0.001
Localized	Ref.		Ref.		Ref.	
Regional	5.08 (2.73–9.44)		5.05 (3.89–6.55)		6.57 (6.08–7.09)	
Distant	18.54 (9.02–38.11)		18.24 (13.54–24.55)		26.75 (24.20–29.56)	
Grade		0.001		<0.001		<0.001
Grade I/II	Ref.		Ref.		Ref.	
Grade III	10.57 (2.92–38.28)		4.31 (2.68–6.94)		3.14 (2.74–3.59)	
Grade IV	6.90 (1.97–24.15)		3.73 (2.37–5.86)		3.01 (2.65–3.42)	
Not Graded	2.83 (0.73–10.90)		2.24 (1.34–3.66)		1.90 (1.64–2.20)	
Surgery		0.013		<0.001		<0.001
Surgery	Ref.		Ref.		Ref.	
No Surgery	3.08 (1.27–7.47)		2.02 (1.44–2.83)		1.90 (1.67–2.17)	

^a^ Hazard ratio; ^b^ Confidence interval; ^c^ Reference group; ^d^ Asian/Pacific islander.

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
