# Peer review of "Does Health Insurance Modify the Association Between Race and Cancer-Specific Survival in Patients with Urinary Bladder Malignancy in the U.S.?"

_ijerph, 2019, doi:10.3390/ijerph16183393_

Round 1

Reviewer 1 Report

The paper by Morales and colleagues describes a statistical analysis on the influence of insurance in the association between race and 5 years urinary bladder cancer survival. It is well known the influence of the race in the survival rates (Wang et al 2018. Sci Reports).

Overall the paper meets the formal structure of a scientific manuscript, and the English is well written and comprehensible.

The principal drawback of the paper is on the interpretation of the results. The main conclusion of the authors:” association between race and 5-year survival depends on insurance status” but looking into Table 1 I am not able to see this association. The HR for uninsured and insured shows two divergent directions in the occurrence of survival, and these heterogeneities depends on the race too. This heterogeneity prevents delivering strong conclusions on the associations between the proposed variables.

Moreover, in the paper from Wang et al, cited by the authors, it is stated that “SEER does not collect data regarding insurance type, educational level, socioeconomic status, availability to healthcare and other factors that affect one’s treatment, and as a consequence, the survival outcome”. Please clarify this point.

Insurance depends on multiple factors not related to the race, and as authors state, there are inequalities on the access to quality care. Other factors influencing 5-year survival (genetic factors, lifestyle, etc..) are not taken into account in these analysis nor stated in the limitations.

However, the study is interesting and if authors reformulate their conclusions can be of interest for the scientific audience of the journal.

Other major comments and recommendations are:

-Describe the differences between being insured and not being insured. Moreover, what are the covered/uncovered services for Medicaid/other insurances. Readers of the paper may not be familiar with the US health insurance system. Please provide background data to understand the proportion of individuals for each race who have an insurance.

- Include the references before the full stop.

Author Response

Dear reviewer, thank you very much for your valuable comments how to improve our manuscript. We have carefully studied each observation/suggestion and marked all changes in the text in yellow.

Comment #1: The principal drawback of the paper is on the interpretation of the results. The main conclusion of the authors:” association between race and 5-year survival depends on insurance status” but looking into Table 1 I am not able to see this association. The HR for uninsured and insured shows two divergent directions in the occurrence of survival, and these heterogeneities depends on the race too. This heterogeneity prevents delivering strong conclusions on the associations between the proposed variables.

Response#1: Table 1 presents the baseline characteristics of the study population according to race. It does not test for associations (or interactions) but checks whether the predictors are differently distributed according to race. The table also helped us to identify possible confounders. Thus, the p-value in this table corresponds to a chi-square test. We have now clarified this in a footnote. In addition we have now added information on insurance status according to race to the table 1.

Effect modification (or interaction) by health insurance was assessed in the logistic regression models by adding the interaction term race*insurance. At least one of the categories for the interaction term of race*insurance was statistically significant in these models showing possible effect modification by health insurance (we have now added a new Table 2 that includes the interaction term as requested as well by reviewer#2). Therefore, the adjusted model was stratified for each stratum of health insurance (Table 3).

Following sentence was added to the description of the statistical analysis section to clarify this better:

The adjusted model with an interaction term for race and insurance status was fitted. As at least one of the interaction terms of race*insurance was statistically significant, showing possible effect modification by health insurance, the adjusted model was stratified for each stratum of health insurance.

In addition, we added following sentence to the results section of Table 2:

The interaction of race*insurance was statistically significant in the adjusted logistic regression models revealing possible effect modification by health insurance. Therefore, the adjusted model was stratified according to health insurance.

Comment #2: Moreover, in the paper from Wang et al, cited by the authors, it is stated that “SEER does not collect data regarding insurance type, educational level, socioeconomic status, availability to healthcare and other factors that affect one’s treatment, and as a consequence, the survival outcome”. Please clarify this point.

Response #2: The reviewer is partly correct as prior to 2007, SEER did not collect information on health insurance of the patients. However, this information is available starting from 2007. This is why our analysis only included study participants from 2007 onwards.

Following sentence was added to section 2.2 to clarify this issue:

The period of this study was chosen due to the SEER database beginning to record insurance status in 2007.

Comment #3: Insurance depends on multiple factors not related to the race, and as authors state, there are inequalities on the access to quality care. Other factors influencing 5-year survival (genetic factors, lifestyle, etc..) are not taken into account in these analysis nor stated in the limitations.

Response #3: We agree with the reviewer that other factors we could not control for may have influenced the association between race and survival. We have added the following information into the limitations of the study section according to your suggestions:

The SEER database does not record detailed information about significant lifestyle habits such as smoking status, occupational exposures and educational status. Also, information on genetic factors were not available. Thus, these factors may over or underestimate the strength of the association between race, health insurance and survival.

Comment #4: Describe the differences between being insured and not being insured. Moreover, what are the covered/uncovered services for Medicaid/other insurances? Readers of the paper may not be familiar with the US health insurance system.

Response #4: We have added following information of the US health-insurance system to offer a better understanding of our system to the readers outside the US and to offer a better description of the insurance categories.  

Healthcare coverage in the U.S. is provided through a combination of private health insurance and public health coverage such as Medicaid.  It has been estimated that close to 91% percent of the U.S. population had some type of health insurance. Most Americans with private health insurance receive it through an employer-sponsored program. Medicaid in the United States is a federal and state program that helps with medical costs for some people with limited income and resources. People who do not have a health-insurance (the uninsured) have to pay for their health services.

SEER report insurance status of the patient based on the primary payer at diagnosis according to NAACCR Field Primary Payer at diagnosis. SEER does not provide much detail about the services covered by Medicaid or other insurances.

Comment #5: Please provide background data to understand the proportion of individuals for each race who have an insurance.

Response#5: We have now added the proportions of the type of health insurance to Table 1 to provide more information ho health insurance was distributed according to race.

Reviewer 2 Report

This manuscript covers a topic that may potentially be of interest to IHERPH readers, and one that is of importance to the literature: the interaction of race and health insurance status in cancer survival.  This manuscript specifically examines bladder cancer cause-specific survival. There is a lot of potential in this paper, but also a lot of revisions required.  I have outlined some general and specific comments, below.

General comments:

1.       The manuscript needs a thorough review for spelling, grammar, and flow. For example, the title is grammatically incorrect, and there are many places throughout the text that the language is awkward. If the authors are not native English speakers, suggest asking a native English speaker to review.

2.       It is hard to decipher whether some null findings are truly null, or whether there is a lack of statistical power.

Specific comments:

1.       Abstract.  The background section of the abstract gives no background, just an objective.  Please revise.

2.       Abstract: It is unclear how the results answer the question posed in the title.

3.       Abstract: I am not sure if I would call this a retrospective cohort study. Rather I would just say “We used data from the SEER 9 database…” etc.

4.       Abstract: The abstract does not make it clear how effect measure modification (interaction) is assessed.

5.       Introduction, line 45. 1% is not a particularly notable decrease.  When the authors say “5 year mortality,” do they average annual mortality rates or survival?  The numbers cited indicate survival statistics are being cited, not mortality rates.

6.       Introduction needs to be reorganized to improve flow.  It currently jumps back and forth between discussions of race and insurance status.  There is little to no discussion of API, who also feature in the present analysis. Also, I feel it’s missing a discussion of insurance status and survival.

7.       Methods: There are a lot of gaps in this section. Questions include:

a.       I assume the SEER 9 database was used from the description? Please specify in text.

b.       What years of data were included? Please specify.

c.       Line 79 – do the authors mean that those missing data on cause of death and death certificate only cases were excluded?

d.       Were cases restricted to first primary cases, and histologically confirmed cases, in accordance with prevailing standards in cancer surveillance research?

e.       Line 82-83. The definition of primary outcome pertains

f.        Were individuals of “other” race excluded?

g.       Lines 92-93 mentions exclusions. These should be described all together in section 2.2.

h.       The variables need more description/information. For example, I assume that the SEER Historic Stage A variable was used, but this is not made clear. More information is also required on the grade variable used.

i.         I would like to know what variables were missing data, how much data were missing, and how the authors handled this missing data. For example, I expect that the treatment variables were not 100% complete for all years.

j.         No information is given on the findings of analyses of collinearity, proportional hazards assumption testing.

k.       Follow-up should be described in results. Give mean/median, range.

8.       Results. The results of the interaction analyses should be presented in addition to the stratified analyses.

9.       Results. The figures should be combined into one with three panels, or preferably, one figure where survival by race and insurance status can be examined together.

10.   Tables.  The number of people in each category/strata would be useful given the wide confidence intervals in some strata

11.   The discussion is weak – it does not do a good job of putting the present results in the context of the literature.  It’s also not clear when the authors are discussing the results of the present study versus those in the literature.  When discussing other published studies, very little information is given about the populations of study, which makes it difficult for the reader to contextualize.

12.   Lines 192-196.  The authors discuss many patient related characteristics in this part of the paragraph, but fail to discuss any provider or healthcare-centered reasons for lack of accessing medical care, such as racial microaggressions or differential treatment of patients by race. The authors might look into this literature.

13.   Could the authors please discuss the potential for residual confounding around income and SES? The argument that insurance status acts as a proxy for income, SES and education is inadequate.

14.   Line 193. Do the authors mean insured or injured?

15.   The authors mention the lack of exposure data several times in the limitation section, which is redundant. However, they also don’t explain why the specifically discussed factors are important.  It is assumed that they relate to bladder cancer risk, but not explicitly stated.

Author Response

Dear reviewer, thank you very much for your valuable comments how to improve our manuscript. We have carefully studied each observation/suggestion and marked all changes in the text in yellow.

Comment #1: The manuscript needs a thorough review for spelling, grammar, and flow. For example, the title is grammatically incorrect, and there are many places throughout the text that the language is awkward. If the authors are not native English speakers, suggest asking a native English speaker to review.

Response#1: The manuscript has now been checked for spelling, grammar, and flow by a native English speaker to comply with the comments of the reviewer.

Comment #2: It is hard to decipher whether some null findings are truly null, or whether there is a lack of statistical power.

Response#2: We agree with the reviewer and have clarified this by adding the following sentence to the limitations section of the manuscript:

The post-hoc power calculations and small sample size of the uninsured in the sample revealed low a power to detect a statistically significant results when comparing Black or API with White individuals among uninsured. Thus, one of the reasons why our data did not reveal a statistically significant associations between race and survival may be due to lack of power.

Comment #3: Abstract.  The background section of the abstract gives no background, just an objective.  Please revise.

Response#3:  We agree. We have added following sentence trying to keep in mind the word limitations for the abstract:

Scientific evidence on the effect of health insurance on racial disparities in urinary bladder cancer patients’ survival is scant.

Comment #4: Abstract: It is unclear how the results answer the question posed in the title.

Response#4:   We have re-structured the results section of the abstract to answer the research question and the title of the study better. The results section of the manuscript reads now as follows:

Results: Health insurance was a statistically significant effect modifier of the association between race and survival.  Whereas, API had a lower hazard of death among the patients with Medicaid insurance (HR 0.67; 95% CI 0.48-0.94 compared with White patients, no differences in survival was found between Black and White urinary bladder carcinoma patients (HR 1.24; 95% CI 0.95-1.61). Among the insured study participants, Blacks were 1.46 times more likely than Whites to die of bladder cancer during the 5-year follow-up (95% CI 1.30-1.64).

Comment #5: Abstract: I am not sure if I would call this a retrospective cohort study. Rather I would just say “We used data from the SEER 9 database…” etc.

Response#5: We have revised the sentence in the abstract that states now as follows:

The database of the cancer surveillance program of the National Cancer Institute’s 2015 (n=39,587) was used.

Comment #6: Abstract: The abstract does not make it clear how effect measure modification (interaction) is assessed.

Response#6: We have added the following sentence to the abstract to explain briefly how interaction was assessed:

An adjusted model with an interaction term for race and insurance status was computed.

Comment #7: Introduction, line 45. 1% is not a particularly notable decrease.  When the authors say “5 year mortality,” do they average annual mortality rates or survival?  The numbers cited indicate survival statistics are being cited, not mortality rates.

Response#7: We agree with the reviewer and have corrected the sentence as follows:

From the years 1989 to 2013, the five year survival rates have remained relatively stable. [1]

Moreover, during 1987-1989 and 2007-2013 five year survival in Whites and Blacks were different indicating health disparity between Whites and Blacks. [1,2]

Comment #8: Introduction needs to be reorganized to improve flow.  It currently jumps back and forth between discussions of race and insurance status.  There is little to no discussion of API, who also feature in the present analysis. Also, I feel it’s missing a discussion of insurance status and survival.

Response#8: We have rewritten the introduction to improve its flow. In regard the discussion,  we have rewritten the paragraph two of the discussion where we discuss in detail our findings with those of previous studies.

Comment #9: Methods: There are a lot of gaps in this section. Questions include:

I assume the SEER 9 database was used from the description? Please specify in text.

We have specified this now in the text.

What years of data were included? Please specify.

The methods section now states as follows:

… from 2007 to 2015 (n = 39,587). The period of this study was chosen due to the SEER 9 database beginning to record insurance status in 2007.

Line 79 – do the authors mean that those missing data on cause of death and death certificate only cases were excluded?

We have revised the sentence that states now as follows:

Cases with missing data on race unknown cause of death or missing information on survival or duplicate cases were excluded.

Were cases restricted to first primary cases, and histologically confirmed cases, in accordance with prevailing standards in cancer surveillance research?

We have added following sentence to clarify this issue:

Thus, the study population was restricted to first primary and histologically confirmed cases.

Line 82-83. The definition of primary outcome pertains

We have added following sentence to clarify this issue:

Thus, the study population was restricted to first primary and histologically confirmed cases.

Were individuals of “other” race excluded?

As stated on line 79, only individuals reporting their race as white, black or Asian/Pacific islander were included in the analysis. 

Lines 92-93 mentions exclusions. These should be described all together in section 2.2.

All exclusion criteria are now stated in section 2.2

The variables need more description/information. For example, I assume that the SEER Historic Stage A variable was used, but this is not made clear. More information is also required on the grade variable used.

Yes, SEER Historic Stage A was used.

We have revised the sentence as follows:

Grade at diagnosis was categorized in four groups: (1) well or moderately differentiated, (2) poorly differentiated, (3) undifferentiated, and (4) ungraded.

I would like to know what variables were missing data, how much data were missing, and how the authors handled this missing data. For example, I expect that the treatment variables were not 100% complete for all years.

We have added following information to the section 2.2 to clarify this item:

Cases with missing data on race (n = 504), unknown cause of death or missing information on survival (n = 1,071), confirmed or possible diagnosis at autopsy (n = 171) or duplicate cases (n = 176) were excluded. For the rest of the variables in the study, the percentage of missing was less than 8% (status: 7.97%, insurance: 5.30%, stage: 1.23%, surgery: 0.18%).

No information is given on the findings of analyses of collinearity, proportional hazards assumption testing.

We have added the following information to the statistical analysis section:

Collinearity was assessed via pairwise correlations. A Pearson’s correlation coefficient absolute value of 0.67 or higher was used for identifying collinearity among a pair of variables. In the study, no collinearity was found among any of the variables included.

Proportional hazard assumption was assessed graphically.

Follow-up should be described in results. Give mean/median, range.

We have added the following information to the results section:

The range of follow up was between 0 and 60 months. Median survival time could not be computed since more than 50% of individuals were alive at the end of follow up. The mean survival time for the sample was 52.6 months.

Comment #10: Results. The results of the interaction analyses should be presented in addition to the stratified analyses.

Response#10: We have now added our working table with the adjusted hazard ratios of the statistical model that included the interaction term (new Table 2) and added a description of the results in the text.

Comment #11: Results. The figures should be combined into one with three panels, or preferably, one figure where survival by race and insurance status can be examined together.

Response#11: As suggest by the reviewer, we have now replaced Figures 1-3 by one figure where survival by race and insurance status was examined together.

Comment #12: Tables.  The number of people in each category/strata would be useful given the wide confidence intervals in some strata

Response#12: We have added this information to Table 2.

Comment #13: The discussion is weak – it does not do a good job of putting the present results in the context of the literature.  It’s also not clear when the authors are discussing the results of the present study versus those in the literature.  When discussing other published studies, very little information is given about the populations of study, which makes it difficult for the reader to contextualize.

Response#13: We have rewritten the paragraph two, three and four of the discussion where we discuss our findings with those of previous studies. Paragraph two now discusses now our results in regard health insurance as effect modifier of the associations between race and survival. Paragraph three now compares the results of our study in regarding the associations between race and survival with previous scientific evidence. We have added a paragraph four that discusses all aspects of possible reasons for the observed findings in order to better structure the discussion. We have also made sure to include more information on the study populations of the previous studies.

Comment #14: Lines 192-196.  The authors discuss many patient related characteristics in this part of the paragraph, but fail to discuss any provider or healthcare-centered reasons for lack of accessing medical care, such as racial microaggressions or differential treatment of patients by race. The authors might look into this literature.

Response#14: We have added a paragraph four that discusses all aspects of possible reasons for the observed findings in order to better structure the discussion. We have also made sure to provide more information on the study populations of the previous studies.

Comment #15: Could the authors please discuss the potential for residual confounding around income and SES? The argument that insurance status acts as a proxy for income, SES and education is inadequate.

Response#15: We have removed the sentence from the discussion as the argument that insurance status acts as a proxy for income, SES and education is too weak.

This statement has more to do with the discussion section and the rationale of using insurance as a proxy for SES. Maybe there is a paper cited in the lit review that can help to argument the choice.

Comment #16: Line 193. Do the authors mean insured or injured?

Response#16: We mean insured. We have corrected that error.

Comment #17: The authors mention the lack of exposure data several times in the limitation section, which is redundant. However, they also don’t explain why the specifically discussed factors are important.  It is assumed that they relate to bladder cancer risk, but not explicitly stated.

Response#17: We have now better summarized that paragraph and added some more details. The following element has now been added to the limitation section:

The SEER database does not record detailed information about significant lifestyle habits such as smoking status, occupational exposures and educational status. Also, information on genetic factors were not available. Thus, these factors may over or underestimate the strength of the association between race, health insurance and survival.

Round 2

Reviewer 1 Report

Authors have answered to all the first round review points. i have some minor considerations:

-Add the period of analysis in the abstract (2007-2015)

-Include the references before the full stop (line 49: relatively stable [1].)

-Consider dividing Figure 1 intro three sub-figures (each for a race). It is really hard to see a difference between race and insurance status.

Author Response

Comment#1: Add the period of analysis in the abstract (2007-2015)

Response#1: We have now added the period of the study to the abstract

Comment#2: Include the references before the full stop (line 49: relatively stable [1].)

Response#2: We agree that it may look better to include the reference before the full stop. However, the guidelines for authors of the journal asks to include the reference after the full stop.

Comment#3: Consider dividing Figure 1 intro three sub-figures. It is really hard to see a difference between race and insurance status.

Response#3: We agree with the reviewer. We have not split the Figures into three new figures and present the Kaplan-Meier curves according to health insurance (our effect modifier). Presenting the figures according to health insurance is now also in line with our research objective and question which was to assess whether health insurance modifies the association between race and survival.